# Sleep, Diet, Physical Activity, and Stress during the COVID-19 Pandemic: A Qualitative Analysis

**DOI:** 10.3390/bs12030066

**Published:** 2022-03-02

**Authors:** Kyanna Orr, Zachary Ta, Kimberley Shoaf, Tanya M. Halliday, Selene Tobin, Kelly Glazer Baron

**Affiliations:** 1Division of Public Health, Department of Family and Preventive Medicine, University of Utah, 375 Chipeta Way A, Salt Lake City, UT 84108, USA; u1045145@utah.edu (K.O.); u0847565@utah.edu (Z.T.); kimberley.shoaf@utah.edu (K.S.); 2Department of Health and Kinesiology, College of Health, University of Utah, 250 S. 1850 E., Salt Lake City, UT 84112, USA; tanya.halliday@utah.edu (T.M.H.); selene.tobin@utah.edu (S.T.)

**Keywords:** COVID-19, diet, physical activity, sleep, stress

## Abstract

The COVID-19 pandemic has changed routines and habits, raising stress and anxiety levels of individuals worldwide. The goal of this qualitative study was to advance the understanding of how pandemic-related changes affected sleep, diet, physical activity (PA), and stress among adults. We conducted semi-structured, qualitative interviews with 185 participants and selected 33 interviews from a represented sample based on age, race, and gender for coding and analysis of themes. After coding for thematic analysis, results demonstrated four primary themes: sleep, diet, PA, and stress. Sleep sub-themes such as poorer sleep quality were reported by 36% of our participants, and 12% reported increased an frequency of vivid dreams and nightmares. PA was decreased in 52% of our participants, while 33% experienced an increase and 15% experienced no change in PA. Participants also reported having an improved diet, mostly among women. Stress was elevated in 79% of our participants and was more likely to be discussed by women. Many participants reported how stress was involved in precipitating health behavior change, especially for sleep. Increased stress was also linked to elevated anxiety and depression among participants. The results of this qualitative study demonstrate how managing stress could have a beneficial effect on promoting health behaviors and mental health during the COVID-19 pandemic and beyond.

## 1. Introduction

The COVID-19 pandemic is the most significant public health emergency affecting the entire world in the past 100 years. The subsequent lifestyle changes due to lockdowns, closures of work and school settings, and social distancing [1,2] significantly disrupted basic health-promotion activities, including sleep, diet, and physical activity (PA). For example, a study in the Netherlands reported that 20% of adults who described themselves as “good sleepers” before the pandemic experienced disrupted sleep during the COVID-19 lockdown [3]. Another study conducted in Italy showed that although 15% of participants reported pre-pandemic difficulties initiating sleep, this rose to 42% during the lockdown period [4]. In addition, 48.6% of a total of 3533 respondents from Italy self-reported weight gain during the pandemic [5]. Furthermore, a longitudinal study in the United States has shown that 40% of 764 participants reported gaining either 1–4 lbs or >5 lbs of body weight during peak lockdown [6]. Studies have also demonstrated reduced PA [7,8]. For instance, in Spain, one survey reported that participants experienced a 12.8% decrease in PA during the COVID-19 lockdown [8]. Moreover, a study in Brazil reported that prior to the COVID-19 pandemic period, 69% of participants were classified as “very active”, while during the social distancing period, this percentage dropped to 39% [9]. A cross-sectional analysis in Italy showed that approximately half of their enrolled sample of 384 had decreased PA [10].

In addition to promoting physical wellness, health behaviors are important for promoting emotional wellness. For example, PA improves stress tolerance [11]. Additionally, incorporating a healthy diet and exercise routine is correlated to stress reduction [12] while good sleep quality is associated with lower stress levels [13]. Health behaviors may also be cumulative, such as poor sleep quality leading to poorer dietary habits and reduced motivation to participate in PA [14]. Dietary changes (e.g., unhealthy diet and overeating) during the COVID-19 pandemic may have a bidirectional relationship with psychological stress. For example, even in periods outside of the pandemic, stress ratings have been associated with increased intake of fat and processed carbohydrates [15,16]. Therefore, the COVID-19 pandemic provides an opportunity to understand the role of stress on health behaviors and consider ways to enhance physical and mental health resilience during times of stress.

Although a substantial number of surveys about health behaviors during the pandemic have been published, few studies have examined the qualitative experiences of the pandemic. This suggests there is a gap in the understanding of detailed, personal experiences with COVID-19, particularly in relation to health and stress. Existing studies tend to have focused on one behavior at a time, rather than the interrelationship between stress and health behaviors. One qualitative analysis was conducted in Canada, where researchers investigated the impact of the pandemic on PA and sedentary behavior. The study found that workout routines were negatively impacted due to lack of access to fitness, sport, and recreational facilities. Sedentary behavior was increased as participants used their time to watch more television during the pandemic [17]. Another project was conducted in London, where researchers utilized qualitative measures to survey people experiencing difficulty sleeping during the pandemic [18]. The study found that participants experienced exacerbated sleep issues, and largely attributed this to the closure of support services and day centers, social isolation, and loss of income. In summary, these studies suggest that more qualitative research is needed to understand the impact of the pandemic on multiple health behaviors.

Accordingly, the goal of this study was to advance the understanding of how the COVID-19 pandemic affected multiple health behaviors (sleep, diet, and PA) and their interrelationship with stress. We aimed to analyze age and gender differences, as well as any differences reported during different months of the pandemic. We conducted a qualitative analysis to delve more deeply into individuals’ experiences with changes in health behaviors during the ongoing pandemic, and to understand factors that were perceived to influence those changes. We hypothesized that there would be major disruptions to health behaviors due to the pandemic, and that these changes will be reflected in reported stress. Results of this qualitative study may also be relevant for analyzing health behavior changes during other stress-inducing events and can provide foundational data for future research and interventions to prevent negative changes in health and wellness.

## 2. Materials and Methods

Data were derived from a mixed-methods study of sleep, diet, PA, and stress level changes during the COVID-19 pandemic. The protocol for this study was approved by the University of Utah Institutional Review Board (IRB_00131837) and all participants were provided with an electronic consent form and provided verbal informed consent.

### 2.1. Participants

A total of 205 adults (age > 18) were recruited for this study and 185 completed the structured interview. A total of 9.8% of the 205 enrolled participants could not be included in this study because they did not complete the required semi-structured interview following enrollment. The study recruitment strategy was planned to enroll a distribution of age and gender among age categories (18–30, 31–45, 45–59, 60+). The inclusion criteria were broad (Utah residence, ability to read and write in English, ability to access study surveys via smartphone, tablet or computer, and willingness to complete the surveys for 3 and 7 day periods). Participants with severe or unstable illnesses that would interfere with participation were excluded (e.g., hospitalization in the past 30 days, current chemotherapy, dialysis, schizophrenia, and dementia).

### 2.2. Procedure

Participants were recruited using online sources, including Research Match, Craig’s List, and Facebook. Interested participants completed a brief preliminary survey to assess eligibility. After completion of the prescreening survey, research staff called interested participants, reviewed the online consent cover letter, and documented verbal, informed consent. Once enrolled, participants were scheduled to complete a 15 min semi-structured interview (Appendix A) scheduled at enrollment. These enrollment interviews were conducted from May to November 2020, and included questions about stress, sleep, diet, and PA-related topics. Utah policy makers lifted restrictions that were placed during the mandatory stay-at-home order in mid-March 2020. During this period, PA was authorized outside, and people were allowed to dine in restaurants, exercise in gyms, and gather up to 20 people while allowing for six feet of social distance and mask wearing. However, professional, or collegiate sporting events, such as basketball and football, prohibited fans from attending games until 2021.

### 2.3. Interview Guide

The interview guide was developed by the lead authors of this study (KGB and KS) and reviewed/edited by several study authors (ZT, KO, TH, and ST). Our semi-structured interview consisted of multiple, open-ended follow-up questions to elicit responses from participants about changes in health behaviors during the pandemic.

### 2.4. Data Analysis

Interviews were transcribed using an automated software (Trint), then manually edited and verified by research assistants. Transcripts were analyzed using inductive and thematic analysis to identify and evaluate themes reported by our participants [19] using an online coding software (Dedoose). Transcripts were coded by two primary coders (KO and ZT) and codes were reviewed and edited by a consensus of three co-authors (KO, ZT, and KB) to ensure intercoder reliability. Given the structure of the interview, we began with a framework of main themes for each health behavior (sleep, diet, PA, and stress) and then identified sub-themes for each of these main health topics. We coded a sample of 33 transcripts that were selected to represent the gender, age distribution, and race/ethnicity of the full sample. Themes were evaluated by age, gender, and month of interview. After analyzing 33 transcripts, no new themes were being generated. Therefore, we discontinued coding as we reached saturation of themes with 33 participants. In qualitative research, the saturation point determines the sample size, as it indicates adequate data have been collected for detailed analysis [20].

## 3. Results

### 3.1. Participants

Demographic information is listed in Table 1. The average age of our sample was 38 years, ranging from 19 to 65 years. A total of 48.5% of the coded interviews were reported by women, 76% of participants were White, and 12% were Hispanic/Latino. Nearly half of the interviews were conducted in the period May–June, 33.4% in the period July–August, and 21.2% in the period September–November. Compared to the full sample of 205 participants, our sample had a similar age and gender distribution.

### 3.2. Sleep

We identified seven themes relevant to sleep (Table 2). The most reported theme was difficulty sleeping, as it was mentioned 17 times over the course of our data collection. Overall, most participants expressed negative changes in their sleep due to the pandemic such as difficulty sleeping and sleeping less. Another sub-theme under sleep changes was increased vivid dreams and nightmares due to the pandemic, reported by 12% of participants. Of those who reported more dreams and nightmares, 75% linked this change to increased stress and anxiety. When discussing changes in their sleep, 12 excerpts discussed sleeping less, 7 discussed sleep quantity “stayed the same” and 6 referred to sleeping more. In summary, more participants reported negative changes in their sleep (poorer sleep quality, less sleep, more dreaming) because of stressors in the pandemic in comparison to positive changes.

We found differences in sleep responses based on gender and pandemic month. Women were more likely to report negative sleep changes during the pandemic in comparison to men. For subcategories “difficulty sleeping”, “sleeping less” and “poor sleep due to stress”, 70%, 75% and 88% of participants reporting these negative sleep behaviors were women, respectively. This contrasts to 30%, 25% and 12% male reports for these same respective categories of sleep. In addition, all of the participants reporting increased dreams and nightmares were women and were nearly all from May 2020. In contrast, more men reported their sleep “stayed the same” throughout the pandemic. Perceptions of sleep based on age were mostly balanced, although half of the participants who reported increased vivid dreams and nightmares were 60 and older.

### 3.3. Diet

We identified eight themes relevant to diet (Table 3). Some participants reported healthier eating habits during the pandemic. For example, 11 excerpts referred to improved eating habits such as eating healthier foods and snacking less frequently. Among these participants, some reported they implemented healthier eating to boost their health and immunity in order to prevent themselves from contracting the virus. Cooking at home was frequently mentioned (23 excerpts) as well as increased snacking (17 excerpts). While positive changes were mentioned among many participants, some negative changes included less healthy eating (8 excerpts) and undesired weight gain (8 excerpts). A total of 21% of participants from our final sample reported they gained weight over the pandemic. While alcohol consumption-based questions in the interview guide were not specifically included, some participants discussed an increase in alcohol consumption (4 excerpts). Overall, both positive and negative changes in diet were reported among participants, with slightly higher prevalence of positive changes.

In analyzing gender, age, and interview month differences, more reports expressing “improved eating habits” came from women (82%) in comparison to men (18%). Most of the participants reporting this change were interviewed between May and June 2020. As such, the majority of participants who reported “weight gain” were men (75%). All of the participants from our sample reporting increased alcohol consumption (4 out of 33) were women and were nearly all reported during May 2020. Similar to sleep, diet changes based on age were mostly balanced except for the subcategory of “more snacking”, which was mostly reported by participants aged 45 and under (82%).

### 3.4. Physical Activity (PA)

We identified three themes relevant to PA (Table 4). The majority of excerpts (25) discussed decreased PA during the pandemic. On the other hand, 12 excerpts discussed increased activity during the pandemic. Some participants reported their decrease in PA was attributed to social distancing, closure of gyms, or avoidance of gyms. Participants who had increases in PA reported having more time for exercise as a result of less time commuting.

For these changes in PA, more than half of participants who reported a decrease in PA were men. The opposite trend was found in women, as 83% of participants who reported “increased activity” during the pandemic were women. These changes in activity levels were mostly reported among participants aged 18 to 30. In addition, most of these changes were reported in the earlier stages of the pandemic (May through July 2020).

### 3.5. Stress

We identified 11 themes relevant to stress (Table 5). The majority of the participants reported increased levels of stress throughout the pandemic. Stressors related to COVID-19 included academic work, reduced socialization and physical interactions, and financial issues. Family stress was also commonly discussed. Participants who were interviewed toward the end of our study attributed their reduced stress levels to adjustment to stress over time. From the 33 participants, there were 19 excerpts that discussed elevated levels of anxiety and 10 that discussed depression.

Results from analyzing stress perceptions based on different age and gender distributions indicate that more women experienced elevated stress during the pandemic in comparison to men. A total of 70% of reports on “stress linked to COVID-19” and 67% of reports on increased “family stress” were from women. On the other hand, 86% of participants who reported “reduced stress” during the pandemic were men. Comparatively more women also expressed increased financial stress and greater anxiety and depression. Many of the participants who reported greater anxiety and depression were between the ages of 46 and 59 (45%) and were predominantly interviewed during the earlier months of the pandemic (May and June). All of the participants from our sample who reported increased “academic stress” were men and were mostly between the ages of 18 and 30. Overall, participants who were interviewed at the beginning of the pandemic (May–July 2020) reported higher stress levels than participants who were interviewed in the later months.

## 4. Discussion

Relevant results of this study demonstrate that most individuals reported substantial changes to sleep, diet, PA, and stress. Few participants reported no change, but it is important to observe that not all changes were negative. Some participants reported improvement in their diet and PA during the pandemic. These changes were attributed to being home more to engage in healthy cooking or having increased free time to exercise due to decreased commute times. These primary results as well as other detailed accounts of health and stress during the pandemic relative to pre-pandemic health provide greater insight into individuals’ experiences, health behaviors, and stress levels during the COVID-19 pandemic.

Results for sleep demonstrated that self-reported sleep quality was compromised among most participants and often linked to increased stress from the COVID-19 pandemic. These negative changes in sleep were largely reported by women while more men reported their sleep quality stayed the same compared to pre-pandemic times. These qualitative results are consistent with survey studies demonstrating that women experienced worse sleep than men during the COVID-19 pandemic. For example, a web-based cross-sectional study of 1908 enrolled participants revealed that more women reported bad sleep and decreased quality of sleep compared to men [21]. Previous research using surveys of participants during the lockdown demonstrated that on average, participants reported later sleep onset and wake times, and diminished sleep quality compared to the pre-lockdown period [22]. A research study evaluating changes in sleep quality among 365 patients showed nearly 70% of patients reported at least one sleep difficulty, and that home confinement, female gender, and sleep-disordered breathing were associated with sleep problems [23]. Although there is research demonstrating compromised sleep quality throughout the pandemic, some studies show that sleep quality has improved during the lockdown period. A qualitative study in Canada revealed that the shutdown of schools due to the COVID-19 pandemic led to a 2 h shift in sleep among adolescents, ultimately developing longer sleep duration, improved sleep quality, and less daytime sleepiness compared to students under a regular school-time schedule [24]. An observational study in Colorado showed that sleep duration increased 30 min during weekdays, 24 min during weekends, and regularity of sleep timing improved by 12 min among university students [25]. Our results also demonstrated an increased frequency of self-reported vivid dreams and nightmares related to stress among some participants. These reports were exclusively shared by women in our final sample of 33 participants and were all reported during May 2022. These results are similar to other reported studies. A study in Toronto showed that while 42.2% of their participant pool experienced heightened nightmares overall, women were significantly more likely to report an increase in nightmares during the pandemic [26]. Another study analyzing the pandemic’s influence on the frequency and content of nightmares found that participants who reported greater general COVID-related stress were more likely to have nightmares. Many nightmares were reported to include content related to the pandemic, such as home confinement [27].

While positive dietary habits were reported in our sample including cooking more at home and eating healthier food, many participants reported higher levels of snacking and unhealthy eating. Increased snacking was largely reported by participants aged 45 and under. A possible reason for this could be that more opportunities for snacking during the day were present for younger participants attending school or working from home. Positive eating habits were more commonly reported by women, while more men reported increases in weight during the pandemic. These data are present in other studies as well. A cross-sectional study of 3703 respondents in Sri Lanka reported that men had a greater increase in weight gain (+4.15 kg) than women (+3.29 kg) [28]. While we did not directly ask for changes in alcohol consumption during the interviews, we found that women were more likely to report an increase in alcohol consumption. Our findings pertaining to unhealthy eating are comparable to other studies but appear to be less extreme. For example, a large sample of European countries reported an increase in consumption of ready-made meals, canned foods, alcoholic drinks, and sweet snacks during the pandemic [29]. Another study reported that out of a sample of 120 participants, 22% gained 5–10 pounds during the pandemic [30]. This finding is similar to our results, as 21% of our final sample reported weight gain.

In terms of PA, our results were somewhat consistent with quantitative studies showing decreased PA levels in many participants [9,31] but also demonstrated increases in PA among some participants. For participants that expressed a decrease in their PA, changes were linked to increased alcohol intake, weight gain, greater fatigue, and elevated stress. Factors that were attributed to increased sedentary behavior included the closing of public gyms, self-isolation, and remote work. While most participants expressed a decrease in PA, there was a notable number of excerpts reporting increased PA over the pandemic period. The reported increase in exercise may have been related to increased time for exercise (e.g., no longer commuting to work or school). However, even among these participants who report more time to exercise, by working from home they may still have had an overall decrease in total PA due to loss of occupational PA as a result of remote work. Given that occupational PA is a significant contributor to total activity [32], it is not surprising that remote work was a strong predictor of total activity during the COVID-19 pandemic [33]. Increases in PA levels were reported by more women than men, while decreases in PA levels were reported by more men than women. Fluctuations in activity levels were largely reported in the earlier stages of the pandemic (May–July 2020) This is in line with previous studies that have demonstrated men exhibit decreased PA compared to women. For instance, data shown from a cross-sectional study in Spain revealed that increased sedentary time was reported to have a higher increase in men than in women. Furthermore, men had significantly reduced activities by 8.2%, while women had an increase in activities by 11% during COVID-19 confinement [34].

As expected, stress during the COVID-19 pandemic was a major theme in the interviews and often linked to changes in health behaviors and sleep. Nearly all participants reported elevated stress from COVID-19 and were more commonly reported by women. Participants frequently reported that pandemic stress (e.g., fear of contracting COVID-19) prevented them from leaving the house to exercise, particularly at public gyms. In addition, stress was often linked as the main cause of sleep disruption and heightened anxiety and depression. This could be an explanation as to why more women experienced increased stress, reduced sleep quality, and greater anxiety and depression than men. Elevated stress, anxiety, and depression in women are shown in current research data. A descriptive cross-sectional study in Morocco consisting of 827 enrolled participants showed that female respondents exhibited more depressive and anxious symptoms compared to men [35]. In addition, cross-sectional studies from Italy and Turkey reported that female participants exhibited heightened acute stress, anxiety, and depression in comparison to male participants during the COVID-19 pandemic [36,37]. This was shown again in a recent study in which men generally reported lower levels of stress compared to women when assessed for stress susceptibility during the pandemic [38]. Decreased stress in men may factor into why more men reported “sleep quality stayed the same” relative to pre-pandemic times. Additional links to stress included academic performance, social isolation, financial health, and family relationships. Overall, stress levels were elevated during the initial months of the pandemic (May–June 2020) compared to later months (July–November 2020), as this may be due to the acclimation of pandemic life and stress over time. Further, pandemic restrictions in Utah were mostly lifted after June 2020.

### Strengths and Limitations

The strengths of our study include the in-depth compilation of individuals’ experiences reacting to the COVID-19 pandemic and its effect on critical health behaviors. Additionally, our sample was racially and ethnically diverse (e.g., 12% Black, 6% Asian, 3% Native Hawaiian or Pacific Islander, and 12% Hispanic) and reflective of the Salt Lake County racial/ethnic make-up (2% Black, 4% Asian, 2% Native Hawaiian and Pacific Islander, and 19% Hispanic). However, results are most pertinent to our specific geographic area, and thus may not be generalizable to individuals living in other areas. For example, our state had closures of schools and in-person dining while many workplaces and businesses remained open. In addition, behaviors are self-reported and therefore may be subject to response biases such as social desirability bias. An additional limitation concerns the duration of this study, in which data collection was limited to the months of May–November 2020. Data collected during this period may not be applicable to circumstances beyond these months. Our interview guide did not specifically query alcohol and substance use, or other behavioral problems such as technology addiction or gambling. Finally, the diversity of ages and different months/phases during the COVID-19 pandemic demonstrates the heterogeneity responses. The results of our study are not specific to a particular age group, which could contribute to a loss in the precision of results. While our goal was to demonstrate a broad impact on health behaviors, we recognize the effects of the pandemic may be different based on occupation or age. For example, our results demonstrated prominent gender differences in many themes. Finally, no concrete directionality was determined because this study was qualitative and cross-sectional. However, it is likely there are bidirectional relationships between stress and other health behaviors as shown in the reported results.

## 5. Conclusions and Implications

In summary, our results demonstrate that stress and health behaviors were closely interrelated during the COVID-19 pandemic. Personal experiences during the initial stages of the pandemic were recorded and analyzed, providing important data on health behavior changes in Utah. Results of this qualitative study may be generalized for future stress-inducing events and can be used for interventions to prevent negative changes in health and wellness. The relationship between stress and critical health behaviors presented through our study results demonstrates the need for greater financial investment towards research in these areas to understand in greater detail the negative implications of COVID-19 on sleep, diet, PA, and stress. Given that positive health behaviors play a significant role in mitigating stress, interventions are needed for maintaining these behaviors during stressful times, including the ongoing COVID-related restrictions for this and future pandemics. For the ongoing challenges associated with this pandemic and in the future, it may be useful to consider interventions for maintaining these behaviors during times when individuals must work remotely and limit outdoor activities (e.g., refraining from going to the gym due to potential illness transmission or contraction). These results further demonstrate the need for wide-scale delivery of stress management interventions, especially during future times of widespread crises such as future epidemics or pandemics. This research is additionally important in understanding disparities in health based on age and gender during the pandemic. As women were found to have increased stress, decreased sleep quality, and elevated anxiety and depression, future interventions to increase access to stress, sleep, and mental health resources for women could help prevent these disparities in health during the on-going pandemic and in future stress-inducing events. A similar approach could be used to combat the trend in weight gain and decreased PA for men. In addition to developing these preventative interventions, we should also consider strategies to help individuals continue the positive behaviors they developed during the pandemic, such as cooking at home more and maintaining improved eating habits.

## Figures and Tables

**Table 1 behavsci-12-00066-t001:** Demographics.

Sample Characteristics		
Gender	N	Percentage
Male	16	48.50%
Female	16	48.50%
Transgender	1	3.00%
Race	N	Percentage
White	25	75.80%
Black	4	12.10%
Asian	2	6.00%
Native Hawaiian/Pacific Islander	1	3.00%
Other	1	3.00%
More than one race	0	0%
Ethnicity	N	Percentage
Hispanic/Latino	4	12.10%
Non-Hispanic/Latino	29	87.90%
Age	N	Percentage
18–30	12	36.40%
31–45	13	39.40%
46–59	6	18.20%
60+	2	6.00%
Marital Status	N	Percentage
Married	22	66.70%
Single/Never married	7	21.20%
Divorced	4	12.10%
Interview Month	N	Percentage
May	9	27.30%
June	6	18.20%
July	5	15.20%
August	6	18.20%
September	4	12.10%
October	2	6.10%
November	1	3.00%

Note. Demographics of the full sample (*n* = 185) included 51% women, 77% White, and age M = 43.82 (SD = 15.83) years.

**Table 2 behavsci-12-00066-t002:** Sleep.

Codes	Number of Excerpts	Representative Quotes
Difficulty sleeping	17	*“I don’t sleep well. My sleep is not as strong, so it gets interrupted... I’m wary of any sounds… [my sleep] is not as deep.”*
Sleeping less	12	*“Because of stress, I sleep less. I wake up in the middle of the night.”*
Poor sleep due to stress	8	*“At the beginning of the pandemic… [my] stress was in the 6–7 range… For the first few weeks, my sleep at the beginning of COVID-19 was on a terrible schedule.”*
Stayed the same	7	*“I’ve been pretty good about my sleep. I still get about 8 h of sleep every night, and I don’t have any trouble falling asleep. My sleep has been really good.”*
Dreams and nightmares	6	*“Sometimes I would sleep late, but I do have these bizarre bouts of bad dreams. Then, I think it’s because of stress, but it’s about really scary stuff. I wake up and wonder why on Earth am I worried about that?”*
Sleeping more	6	*“I wake up a lot later than I normally do and go to sleep a lot later than I normally do, so that’s really thrown off for me personally.”*
		*“There could be multiple factors for the fatigue, but it’s definitely hit harder in the last two weeks where I’m just tired.”*
Fatigue and tiredness	6	*“I am gaining more weight and I am more tired because I know I’m not doing anything. Lately, I have just been too busy with some other family stuff I have.”*

**Table 3 behavsci-12-00066-t003:** Diet.

Codes	Number of Excerpts	Representative Quotes
Cooking at Home	23	*“[We’re eating] healthier because we cook more at home, and we’re more conscientious of what we put in our food.”*
More Snacking	17	*“I go to snacks and stuff that are really quickly accessible, and so I eat a little bit more junk food [during the pandemic].”*
Improved Eating Habits	11	*“I’m really paying attention to what I’m eating and I’m able to eat a lot more fruit and vegetables because I have time to prepare food.”* *“I bought lemons and cabbages excessively, like things to boost my immune system … I amped up my immunity with supplements like B complex vitamins, vitamin C. and multivitamins.”*
Weight Gain	8	*“I’m a stress eater, which didn’t help matters and [I’m] not working out as much. I did gain weight [during the pandemic], which was frustrating.”*
Unhealthy Eating	8	*“Being locked up at home... I eat a little more gelato, chocolate and more of all those things. Because I’m always home.”*
Eating Less	5	*“Especially with being home now, my eating habits have lessened… I definitely eat a lot less than I did when I had a regular schedule.”*
Increased Alcohol	4	*“[I am] drinking more wine during the pandemic. A glass of wine 5 times a week [compared to] one like normal.”* *“Being at home all the time has definitely made me more stressed because I can’t work out, and now I think I [have been] drinking more earlier in the day and … drink on the weekends.”*
More Consistent Eating	2	*“Pre-pandemic, after seven o’clock, I didn’t eat. So, it was always hard for me to have that scheduled time to eat my dinner, I guess. [Now], I have a solid breakfast time where I usually eat between 7 to 8 am. And then, I’ll always have a solid lunch time where I’ll always eat around noon. And then, at dinner time, I’ll always [eat] before 7 pm.*

**Table 4 behavsci-12-00066-t004:** Physical Activity.

Codes	Number of Excerpts	Representative Quotes
Less Activity	25	*“I think just being at home all the time and not being able to exercise and see other people has definitely made me more stressed. So, I think I just start drinking [alcohol] earlier in the day and couple of drinks on the weekend to relax a bit.”* *“I’ve been having this back pain for the past couple months that I think has resulted from working from home and sitting at a desk all day... I’m not exercising as much.”*
More Activity	12	*“I’ve been exercising more at home to try to get rid of just like excess energy that I have from being cooped up inside.”* *“I have more time to exercise because I am not sitting in my car for a couple of hours each day commuting to my job.”*
Same as Before	6	*“I like to be in physical shape … My exercise is the same.”*

**Table 5 behavsci-12-00066-t005:** Stress.

Codes	Number of Excerpts	Representative Quotes
Stress Linked to COVID-19	27	*“[My stress has been] a bit higher, but mostly the stress has changed from being stressed from work to being stressed out more about a virus that could potentially kill some members of my family that have immune and autoimmune diseases.”*
Family Stress	21	*“My husband is normally gone 5–6 days a week at work, so having him home most of the time has changed our routine. I am trying to direct kids away from him while he is working, but he is kind of upset with the distractions around and not getting stuff done. He is an anomaly at home, and no one should be around him.”*
Anxiety and Depression	20	*“I’m a therapist and I’m seeing a ton of anxiety. Depression has gone way up also, even for me. Stuff has come back for everybody. Even when they’ve done a lot of work, all the old unhealthy things are coming back.”*
Increased Stress	15	*“I would say it’s gone up a little bit. And like I said, I think the first four to six weeks were pretty chill. We were all having a great time being inside, but now I am stress out because of that. I don’t know. I’m lucky I kept my job, but you never know when that’s going to change.”*
Financial Stress	11	*“It’s really brought my attention to how shitty not only my financial situation is, but also everyone else’s”*
Academic Stress	10	*“I was taking classes in the Spring of this year. A lot of the classes had compromised quality. We didn’t get to do a lot of the content that we would normally have gotten to if it weren’t for the pandemic.”*
Reduced Socialization and Physical Interaction	9	*“I have to admit, I am getting restless. I’d like to see more friends. I’d like to do things with people, but it’s just not a good idea. So, it’s been challenging.”*
Uncertainty and Unknowns	7	*“When [the pandemic] first started, it was very stressful because of the uncertainty and the thoughts of dying.”*
Reduced Stress	7	*“Now, my stress levels have reduced. Before the pandemic, I was an instructor and a student as well. So, most often I was doing homework and preparing for a class. But during the pandemic, all of our classes were moved online, so I could wait and then do things at my own pace. I would say that now my [stress levels] kind of reduced. I’m more relaxed now than before.”*
Feeling “Trapped” or Lonely	6	*“Yeah, because now it’s like you got nowhere to go. You know, you are just in your house doing stuff at home. So, it’s a little bit more stressful than usual…”*
Frustration	5	*“I am frustrated and not satisfied with the trends in terms of the virus. I am just feeling very frustrated and restricted socially. It’s hard to maintain all your friendships when all you have is the phone or Zoom. Friendships rely on people being together and having shared experiences on their own, and it’s very frustrating.”*

## Data Availability

The data presented in this study are available on request through Kelly Glazer Baron (kelly.baron@utah.edu).

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
