# Peer review of "Sleep, Diet, Physical Activity, and Stress during the COVID-19 Pandemic: A Qualitative Analysis"

_behavsci, 2022, doi:10.3390/bs12030066_

Round 1

Reviewer 1 Report

The topic is interesting, relevant and current. The article  can serve as a preliminary study.

Some claims can be further justified with recent quotes.

There is a lack of data on the development of the instrument, its reliability...

The sample seems well selected and sufficient for this type of qualitative studies. However, independent variables are not  taken into account for the presentation of the results.  It would be interesting to see the different perceptions about COVID based on age,  gender...

In a situation as changing as the one analyzed, and in the case of such a local context, the month of the interview can determine the results. It is not clear whether this type of analysis has been done.

References should be revised to conform to the format and incorporate DOI.

Author Response

The topic is interesting, relevant and current. The article  can serve as a preliminary study.

Some claims can be further justified with recent quotes.

Thank you for your time and attention in reviewing this manuscript. We agree that some claims needed further justification, and have gone through the paper inserting more quotes from recent studies to strengthen our claims. 

There is a lack of data on the development of the instrument, its reliability.

We appreciate this comment. We have added to the Materials and Methods section of our paper to include more information on our interview guide and its development. (see line 363). Our interview guide was not standardized and was intended to have open-ended prompts.

The sample seems well selected and sufficient for this type of qualitative studies. However, independent variables are not  taken into account for the presentation of the results.  It would be interesting to see the different perceptions about COVID based on age,  gender...

We have received similar feedback from the other reviewers and have expanded on each of our categories of sleep, diet, PA, and stress to include results across these independent variables within our “Results” section. 

In a situation as changing as the one analyzed, and in the case of such a local context, the month of the interview can determine the results. It is not clear whether this type of analysis has been done.

We agree that including data showing differences in perceptions of COVID based on interview month is important and can add to our results. We completed a new analysis on our participant reports based on individual months between May and November, and have included this throughout our results section for each subsection (sleep, diet, PA, and stress.)

References should be revised to conform to the format and incorporate DOI.

Thank you for bringing this to our attention. We have revised all of our sources to match the correct format and have inserted their DOIs.

Reviewer 2 Report

This is, in summary, an interesting study aimed to advance the understanding of how pandemic-related changes affected sleep, diet, physical activity (PA), and stress among adults conducting a semi-structured, qualitative interviews with 185 participants. The authors reported four primary themes: sleep, diet, PA, and stress. In addition, sleep sub-themes included poorer sleep quality and increased frequency of vivid dreams and nightmares. Moreover, PA decreased in some participants and increased in others. Moreover, participants also reported having an improved diet and stress was elevated among the majority of participants during the pandemic. Finally, many participants reported how stress was involved in precipitating health behavior change, especially for sleep and increased stress was also linked to elevated anxiety and depression among participants.

The authors may find as follows my main comments/suggestions.

First, the Introduction section is really too short as currently presented; thus, it should be developed in a more detailed manner for the general readership.

Importantly, how participants have been recruited is a matter of debate that should be specified by the authors. In addition, the most relevant reasons related to the exclusion of the 9.8% of subjects who really did not complete the structured interview need to be specified extensively.

Furthermore, the authors could immediately present and discuss, in the first lines of the Discussion section, their most relevant study findings. Conversely, they seem to stress the most relevant aims/objectives of the study that might be mentioned elsewhere within the paper.

Importantly, the most relevant study limitations/shortcomings should be described in a more detailed manner as the the main caveats have been only partially reported. 

Finally, what is the take-home message of this paper? While the authors highlighted that stress and health behaviors were closely interrelated during the COVID-19 pandemic, they failed, in my opinion, to focus on some conclusive remarks to this specific regard. Specifically, what are the main implications of this paper according to these results? How the mentioned results may be generalized? Here, some additional details might be useful for the general readership and should be provided by the authors based on their expertise.

Author Response

Introduction

The Introduction section is really too short as currently presented; thus, it should be developed in a more detailed manner for the general readership.

Thank you for your time and attention in reviewing this manuscript. We have added more literature and explanations on why this study is relevant throughout the introduction.

Methods

How participants have been recruited is a matter of debate that should be specified by the authors. In addition, the most relevant reasons related to the exclusion of the 9.8% of subjects who really did not complete the structured interview need to be specified extensively.

We have clarified this section in the “procedure” section of our Materials and Methods (see line 348). We did not have any data available for 9.8% of the recruited subjects because they did not schedule an interview following enrollment, and have inserted an explanation on this in the manuscript (see line 337). 

Discussion

The authors could immediately present and discuss, in the first lines of the Discussion section, their most relevant study findings. Conversely, they seem to stress the most relevant aims/objectives of the study that might be mentioned elsewhere within the paper.

We have rephrased the introductory Discussion paragraph to highlight the relevant results of the study (see line 744).

The most relevant study limitations/shortcomings should be described in a more detailed manner as the the main caveats have been only partially reported. 

Thank you for this suggestion. We have received similar feedback from the other reviewers and have reported in greater detail the relevant limitations of our study.

What is the take-home message of this paper? While the authors highlighted that stress and health behaviors were closely interrelated during the COVID-19 pandemic, they failed, in my opinion, to focus on some conclusive remarks to this specific regard. Specifically, what are the main implications of this paper according to these results? How the mentioned results may be generalized? Here, some additional details might be useful for the general readership and should be provided by the authors based on their expertise.

We appreciate your detailed input on this. We have updated the conclusion to have more comprehensive information on the main implications of the paper, how the results can be generalized, and other interventions that can be developed. 

Reviewer 3 Report

Thank you for the opportunity to review this interesting manuscript. Overall, this paper has a relevant focus. The research is of interest for researchers and clinicians aiming at improvement of physical and mental health specifically during these challenging times. I have some remarks, mainly ideas that in my opinion should be clarified. I hope the authors will find the comments helpful for the further process.

Abstract:

I suggest being more specific regarding the results, saying that “PA was decreased in some participants and increased in others” is vague.

Introduction:

For me, further justification on why you choose this study, the contribution of this findings to the actual state of art and to the gaps in the literature is needed. More specifically, at this moment it is well acknowledged that COVID-19 pandemic changed lifestyle behaviors, namely sleep, diet and PA, therefore what is the innovative evidence that this study will provide? Why is still necessary to study this topic?

PA abbreviation – please be consistent using the PA abbreviation, sometimes you use abbreviation, others the full term.

Materials and Methods:

Regarding sample selection, some ideas need explanation, namely:

Line 70: “A total of 205 adults (age >18) were recruited for the study and 185 completed the structured interview.”

On table 1 only 33 participants were included, why? If you have 185 interviews and analyzed only 33 in my opinion you are losing several important data.

In the same way, the authors stated that “recruitment strategy was planned to enroll a distribution of age and sex among age categories (18-30, 31-45, 45-59, 60+)”. My question is: if it was your strategy to distribute based on age and sex why do you have only 1 “transgender”? Why do you have only 2 aged “60+”?

Please clarify.

Regarding the Procedures, the authors stated that “These enrollment interviews were conducted from May through November of 2020” (line 83). In my opinion it is important to clarify regarding the state of the COVID-19 pandemic in these moments: was the country in lockdown period? Was PA authorized outside? Were the gyms opened? And about sports teams, were practicing as normal?

Accordingly with the government rules and health standards imposed, the answers could have change.

Results

The authors reported the results found overall but no evidence is mentioned concerning the distributed classes according with sex and age. The change in PA behaviours was the same among males and females? Was the same among age categories 18-30 and 60+?

While reporting the data related with stress (line 145) it is very evident the difference in the results accordingly with the period of time/ state of the COVID-19. In my opinion, although this was not reported on PA, sleep or diet topic, represents a bias also regarding these variables.

For me, this is the major limitation of the present study, the diversity of ages (perhaps different occupations e.g., work, school, retirement), and different periods of time/ state of COVID-19, compromises the directionality of the relationships observed.

Discussion and Conclusion

The manuscript will benefit with the use of comparison/contrasting literature; therefore, it is my suggestion to add more comparative studies.

If the authors decide to change the results section, it will be necessary to rephrase the discussion and conclusion sections.

Limitations need expanding in my view; More technical discussion of limitations could be given.

Author Response

Abstract

I suggest being more specific regarding the results, saying that “PA was decreased in some participants and increased in others” is vague.

We appreciate your time in reviewing our manuscript and helping us strengthen it. We agree that more specific data could have been included in the abstract, and have revised it to include less ambiguous results.

Introduction

For me, further justification on why you choose this study, the contribution of this findings to the actual state of art and to the gaps in the literature is needed. More specifically, at this moment it is well acknowledged that COVID-19 pandemic changed lifestyle behaviors, namely sleep, diet and PA, therefore what is the innovative evidence that this study will provide? Why is still necessary to study this topic?

We have made the necessary revisions tailored to your suggestions. We have added more literature, as well as added the relevance of our study.

PA abbreviation – please be consistent using the PA abbreviation, sometimes you use abbreviation, others the full term.

We have gone through the paper and revised all the “physical activity” terms to the PA abbreviation.

Materials and Methods

Regarding sample selection, some ideas need explanation, namely:

Line 70: “A total of 205 adults (age >18) were recruited for the study and 185 completed the structured interview.”

On table 1 only 33 participants were included, why? If you have 185 interviews and analyzed only 33 in my opinion you are losing several important data.

Thank you for commenting on this. We have clarified the reason why 33 transcripts were used for data analysis on line 379 with information on how we reached “saturation of themes.” In qualitative research, saturation is reached when adequate data is reached for detailed analysis. Because no new codes were being generated at 33 transcripts, coding was discontinued and no further transcripts were analyzed.

In the same way, the authors stated that “recruitment strategy was planned to enroll a distribution of age and sex among age categories (18-30, 31-45, 45-59, 60+)”. My question is: if it was your strategy to distribute based on age and sex why do you have only 1 “transgender”? Why do you have only 2 aged “60+”? Please clarify.

Thank you for this question. We used numbers that were representative of our full sample of 205 participants, which is why there are relatively few transcripts from transgender and elderly participants among the 33 final transcripts analyzed.

Regarding the Procedures, the authors stated that “These enrollment interviews were conducted from May through November of 2020” (line 83). In my opinion it is important to clarify regarding the state of the COVID-19 pandemic in these moments: was the country in lockdown period? Was PA authorized outside? Were the gyms opened? And about sports teams, were practicing as normal?

Accordingly with the government rules and health standards imposed, the answers could have change.

We agree that it is important to address the state of the COVID-19 pandemic from May through November of 2020, and have updated our procedures with more information on each of the points you addressed. (See line 355).

Results

The authors reported the results found overall but no evidence is mentioned concerning the distributed classes according with sex and age. The change in PA behaviours was the same among males and females? Was the same among age categories 18-30 and 60+?

We received similar feedback from the other reviewers and have expanded on each of our categories of sleep, diet, PA, and stress to include results across these independent variables within our “Results” section.  

While reporting the data related with stress (line 145) it is very evident the difference in the results accordingly with the period of time/ state of the COVID-19. In my opinion, although this was not reported on PA, sleep or diet topic, represents a bias also regarding these variables.

We agree that the presence of data on the period of time/ state of COVID should not have exclusively been mentioned for stress. We have incorporated more data on differences in perceptions for COVID based on different months during the pandemic for the rest of our health behaviors (sleep, diet, and PA). This information can be found throughout our results section under each subcategory.

For me, this is the major limitation of the present study, the diversity of ages (perhaps different occupations e.g., work, school, retirement), and different periods of time/ state of COVID-19, compromises the directionality of the relationships observed.

We agree that this is a major limitation and have included it in our strengths and limitations (see line 1363). We have additionally reported in greater detail other relevant limitations of our study to expand on this section of our paper.

Discussion and Conclusion

The manuscript will benefit with the use of comparison/contrasting literature; therefore, it is my suggestion to add more comparative studies.

We appreciate this comment. We have revised the discussion section to include more comparative studies. We have additionally included some contrasting studies to make the discussion more informative.

If the authors decide to change the results section, it will be necessary to rephrase the discussion and conclusion sections.

Thank you for including this important reminder. Due to changes made in the results section, we have revised our discussion and conclusion sections to more adequately reflect our comprehensive results.

Limitations need expanding in my view; More technical discussion of limitations could be given.

Thank you for this suggestion. We have received similar feedback from the other reviewers and have reported in greater detail the relevant limitations of our study using a more technical approach.

Round 2

Reviewer 3 Report

Dear Authors,

Thank you for making considerations as requested. Your manuscript improved significantly. Congratulations for your work.

I am happy with the changes made in context to my recommendations.

I wish you the best for your work!